# Characterization of the ABC methionine transporter from *Neisseria meningitidis* reveals that lipidated MetQ is required for interaction

Naima G Sharaf[1,2†]*, Mona Shahgholi[1], Esther Kim[1], Jeffrey Y Lai[1,2], David G VanderVelde[1], Allen T Lee[1,2], Douglas C Rees[1,2]*

[1]Division of Chemistry and Chemical Engineering, California Institute of Technology, Pasadena, United States; [2]Howard Hughes Medical Institute, California Institute of Technology, Pasadena, United States

**Abstract** NmMetQ is a substrate-binding protein (SBP) from *Neisseria meningitidis* that has been identified as a surface-exposed candidate antigen for meningococcal vaccines. However, this location for NmMetQ challenges the prevailing view that SBPs in Gram-negative bacteria are localized to the periplasmic space to promote interaction with their cognate ABC transporter embedded in the bacterial inner membrane. To elucidate the roles of NmMetQ, we characterized NmMetQ with and without its cognate ABC transporter (NmMetNI). Here, we show that NmMetQ is a lipoprotein (lipo-NmMetQ) that binds multiple methionine analogs and stimulates the ATPase activity of NmMetNI. Using single-particle electron cryo-microscopy, we determined the structures of NmMetNI in the presence and absence of lipo-NmMetQ. Based on our data, we propose that NmMetQ tethers to membranes via a lipid anchor and has dual function and localization, playing a role in NmMetNI-mediated transport at the inner membrane and moonlighting on the bacterial surface.

*For correspondence:
ngsharaf@stanford.edu (NGS);
dcrees@caltech.edu (DCR)

Present address: †Department of Biology, Stanford University, Stanford, United States

Competing interests: The authors declare that no competing interests exist.

## Introduction

The substrate-binding protein (SBP) NmMetQ from the human pathogen *Neisseria meningitidis* has been identified as a surface-exposed candidate antigen for the meningococcal vaccine (*Pizza et al., 2000*). Subsequently, NmMetQ has been shown to interact with human brain microvascular endothelial cells (*Kánová et al., 2018*), potentially acting as an adhesin. However, the surface localization of NmMetQ challenges the prevailing view that SBPs reside in the periplasm of Gram-negative bacteria (*Thomas and Tampé, 2020*), binding and delivering molecules to cognate ATP-Binding Cassette (ABC) transporters in the inner membrane (IM). Several questions arise from these studies: Has NmMetQ lost its ABC transporter-dependent function in the IM? How does NmMetQ become embedded in the outer membrane (OM) surface of the bacterium?

The ABC transporter-dependent role of SBPs has been well characterized for multiple ABC transporter systems (*Hollenstein et al., 2007*; *Oldham et al., 2013*; *Sabrialabed et al., 2020*; *Liu et al., 2020*; *Nguyen et al., 2018*; *de Boer et al., 2019*). These studies reveal conserved SBP-dependent characteristics, including that the SBP is largely responsible for substrate delivery to the ABC transporter, with concomitant stimulation of the transport coupled ATPase activity. Structural studies have shown that SBPs dock to the periplasmic surface of the transporter's transmembrane domains, with the substrate-binding pocket juxtaposed with the translocation pathway of the transporter. While many SBPs have only been assigned ABC transporter-dependent functions, a few SBPs have also been shown to have both ABC transporter-dependent and ABC transporter-independent

functions (often referred to as moonlighting functions) (*Adler, 1975*). For example, the *E. coli* maltose SBP (MBP) binds and stimulates its cognate ABC transporter during the maltose import cycle (*Davidson et al., 1992*). In addition, the MBP-maltose complex is also a ligand for the chemotaxis receptor, triggering the signaling cascade involved in nutrient acquisition (*Hazelbauer, 1975*; *Manson et al., 1985*). Other SBPs have also been assigned ABC transporter-independent functions (*Müller et al., 2007*; *Castañeda-Roldán et al., 2006*; *Matthysse et al., 1996*), including NspS from *Vibrio cholerae*, which has been shown to play a role in biofilm formation (*Young et al., 2021*) and not transport (*Cockerell et al., 2014*). Additionally, two MetQ proteins, *N. gonorrhoeae* (Ng) NgMetQ and *Vibrio vulnificus* (Vv) VvMetQ have also been identified as putative adhesins, mediating bacterial adhesion to human cervical epithelial cells (*Semchenko et al., 2017*) and to human intestinal epithelial cells (*Lee et al., 2010*; *Yu et al., 2011*), respectively. Evidence that these MetQ SBPs bind and stimulate their cognate ABC transporters, however, is lacking. Whether NmMetQ has lost its ATP transporter-dependent function or whether it plays roles at both the IM and OM cannot be determined through amino acid sequence alone and must be experimentally verified.

Since SBPs are not membrane proteins, the detection of NmMetQ at the cell surface of the bacterium suggests it must be tethered to the OM. In Gram-negative bacteria, the paradigm that SBPs translocate into the periplasm where they diffuse freely between the IM and OM can be traced back to early experiments by Heppel showing that the osmotic shock of Gram-negative bacteria leads to the release of SBPs (*Heppel, 1969*). While many SBPs in Gram-negative bacteria have been identified as secreted proteins (*Willis and Furlong, 1974*; *Ahlem et al., 1982*), several studies have also identified a few lipid-modified SBPs (lipo-SBP) (*Tokuda et al., 2007*). However, the presence of lipo-SBPs in Gram-negative bacteria has not been generally appreciated and the role that lipid modifications play in SBP surface localization remains unexplored.

Although ABC transporter-dependent functions of NmMetQ, VvMetQ, and NgMetQ are not well studied, the homologous SBP from *E. coli*, EcMetQ, is well characterized. Studies show that the *E. coli* methionine uptake system consists of EcMetQ and its cognate ABC transporter EcMetNI (*Kadner, 1974*; *Kadner, 1977*). Structures of both EcMetQ and EcMetNI alone and in complex are available (*Kadaba et al., 2008*; *Johnson et al., 2012*; *Nguyen et al., 2015*). EcMetNI comprises two transmembrane domains (TMD), which form a substrate translocation pathway, and two nucleotide-binding domains (NBD), which couple transport to the binding and hydrolysis of ATP. In the absence of EcMetQ, EcMetNI adopts the inward-facing conformation, with the TMDs open to the cytoplasm and NBDs separated. The available crystal structures of EcMetQ reveal two domains connected by a linker that form the methionine-binding pocket (*Nguyen et al., 2015*). Of note, EcMetQ has been experimentally verified to be a lipoprotein by radioactive palmitate labeling (*Tokuda et al., 2007*). Additionally, Carlson et al. found that wild-type EcMetQ remains associated with recombinantly expressed his-tagged EcMetNI when solubilized in detergent and peptidiscs but not when its N-terminal cysteine is mutated to prevent lipidation (*Carlson et al., 2019*). This study shows that EcMetQ association with EcMetNI depends on its N-terminal lipid. Structures of EcMetQ are also available, however, the lipid modification is not present in EcMetQ structures. A structure of the EcMetQ: EcMetNI complex is also available and shows EcMetNI in the outward-facing conformation, with the TMDs and NBDs close together. In this structure, EcMetQ is docked to the periplasmic surface of the TMDs with the binding pocket open to the central cavity (*Nguyen et al., 2018*). These structures, together with in vivo functional assays (*Nguyen et al., 2018*; *Kadner, 1974*), show that EcMetQ is intimately involved in EcMetNI-mediated methionine transport.

Although the interaction between EcMetQ and EcMetNI is well characterized, less is known about the corresponding system in *Neisseria meningitidis*. To date, there have been no biochemical or structural studies reported for NmMetNI. Recently determined structures of NmMetQ are in the ligand-free, L-methionine-, or D-methionine-bound states, and binding assays show L-methionine binds NmMetQ with greater affinity than D-methionine (*Nguyen et al., 2019*). These studies were carried out with an NmMetQ protein that lacks the native N-terminal signal sequence, establishing that the N-terminal signal sequence is not necessary for ligand binding. However, NmMetQ is predicted to be lipoprotein based on the N-terminal protein sequence (Uniprot entry Q7DD63) (*UniProt Consortium, 2019*). Experimental evidence confirming this modification, however, has not been reported. Thus, a full understanding of the post-translational modification of NmMetQ and its interactions with NmMetNI are lacking. To better understand NmMetQ and the role it plays in

methionine transport, a detailed characterization of both NmMetNI and NmMetQ with its native N-terminal signal sequence is required.

In this work, we characterized NmMetQ and NmMetNI using multiple biophysical methods. Using mass spectrometry and site-directed mutagenesis, we demonstrate that full-length NmMetQ, recombinantly-expressed in *E. coli*, is a lipoprotein (lipo-NmMetQ). Functional assays showed that both lipo-NmMetQ and L-methionine are required for maximal stimulation of NmMetNI ATPase activity. NmMetNI was also stimulated to a lesser extent by pre-protein NmMetQ (a variant with an unprocessed N-terminal signal peptide) with L-methionine and by lipo-NmMetQ with select methionine analogs. We also determined the structures of NmMetNI in the presence and absence of lipo-NmMetQ to 6.4 Å and 3.3 Å resolution, respectively, using single-particle electron cryo-microscopy (cryo-EM). Using a bioinformatics approach, we also identified MetQ proteins from other Gram-negative bacteria that are predicted to be modified with lipids. This analysis suggests that the lipid modification of MetQ proteins is not restricted to *N. meningitidis* and *E. coli*.

Based on our data, we propose that lipo-NmMetQ, and more generally lipo-MetQ proteins in other Gram-negative bacteria, possesses dual function and localization: ABC transporter-dependent roles at the IM and a moonlighting ABC transporter-independent role (or roles) at the OM. Our findings highlight the complexity of the cell envelope and that much remains to be understood about the rules governing protein localization in Gram-negative bacteria and the moonlighting functions of SBPs on the surface of the cell.

## Results

### *N. meningitidis* MetQ is a lipoprotein

While lipoproteins and secreted proteins both must traverse the inner cell membrane during biogenesis, their maturation occurs through different mechanisms depending on the N-terminal signal sequence (*Figure 1A*). Lipoproteins are synthesized in the cytoplasm as pre-prolipoproteins, inserted into the IM, and then anchored via their N-terminal signal sequence to the cytoplasmic membrane (*Okuda and Tokuda, 2011*). While tethered to the IM through the signal sequence, pre-prolipoproteins are subsequently modified by three enzymes: (1) phosphatidylglycerol transferase (Lgt), which transfers the diacylglycerol group preferentially from phosphatidylglycerol (PG) to the cysteine residue via a thioester bond of the pre-prolipoprotein, producing a prolipoprotein (*Mao et al., 2016*) (2) signal peptidase II (LspA), which cleaves the prolipoprotein N-terminal signal sequence to yield a diacylated lipoprotein with the N-terminal cysteine *Hussain et al., 1982*; *Vogeley et al., 2016*; and (3) apolipoprotein N-acyl transferase (Lnt), which N-acylates the cysteine residue preferentially using an acyl group of phosphatidylethanolamine (PE) to produce a triacylated lipoprotein. (*Noland et al., 2017*; *Wiktor et al., 2017*). Similar to lipoproteins, secreted proteins are synthesized in the cytoplasm as pre-proteins with an N-terminal signal sequence. These pre-proteins serve as substrates for signal peptidase I (Spase I), which cleaves the N-terminal signal sequence to yield the mature secreted protein (*Karla et al., 2005*; *Paetzel et al., 1998*).

NmMetQ is predicted to be a lipoprotein by SignalP 5.0, a deep neural network algorithm that analyzes amino acid sequences to predict the presence and location of cleavage sites (*Almagro Armenteros et al., 2019*). To validate this prediction, we expressed NmMetQ using an *E. coli* expression system with the native N-terminal signal sequence and a C-terminal decahistidine tag. *E. coli* has been previously used to produce lipid-modified *N. meningitidis* proteins (*Fantappiè et al., 2017*). We purified NmMetQ in the detergent n-dodecyl-β-D-maltopyranoside (DDM) using an immobilized nickel affinity column followed by size-exclusion chromatography (SEC). The SEC elution profile shows one main peak with an elution volume of 66 mL (*Figure 1A*). An analysis of the peak fraction by liquid chromatography mass spectrometry (LC/MS) revealed two major deconvoluted masses of 31,662 and 31,682 Da (*Figure 1B*). These masses correspond well with the theoretical masses of two lipoprotein NmMetQ proteins: one with a triacyl chain composition of 16:0, 16:0 and 16:0 (31,661 Da) and another with a triacyl chain composition of 16:0, 16:0 and 18:1 (31,685 Da), respectively (*Figure 1A*, top). We calculated the intact masses of the lipo-NmMetQ proteins using a combination of 16:0 and 18:1 acyl chains because these were the major species found in previous studies of recombinantly expressed lipoproteins (*Hantke and Braun, 1973*; *Luo et al., 2016*).

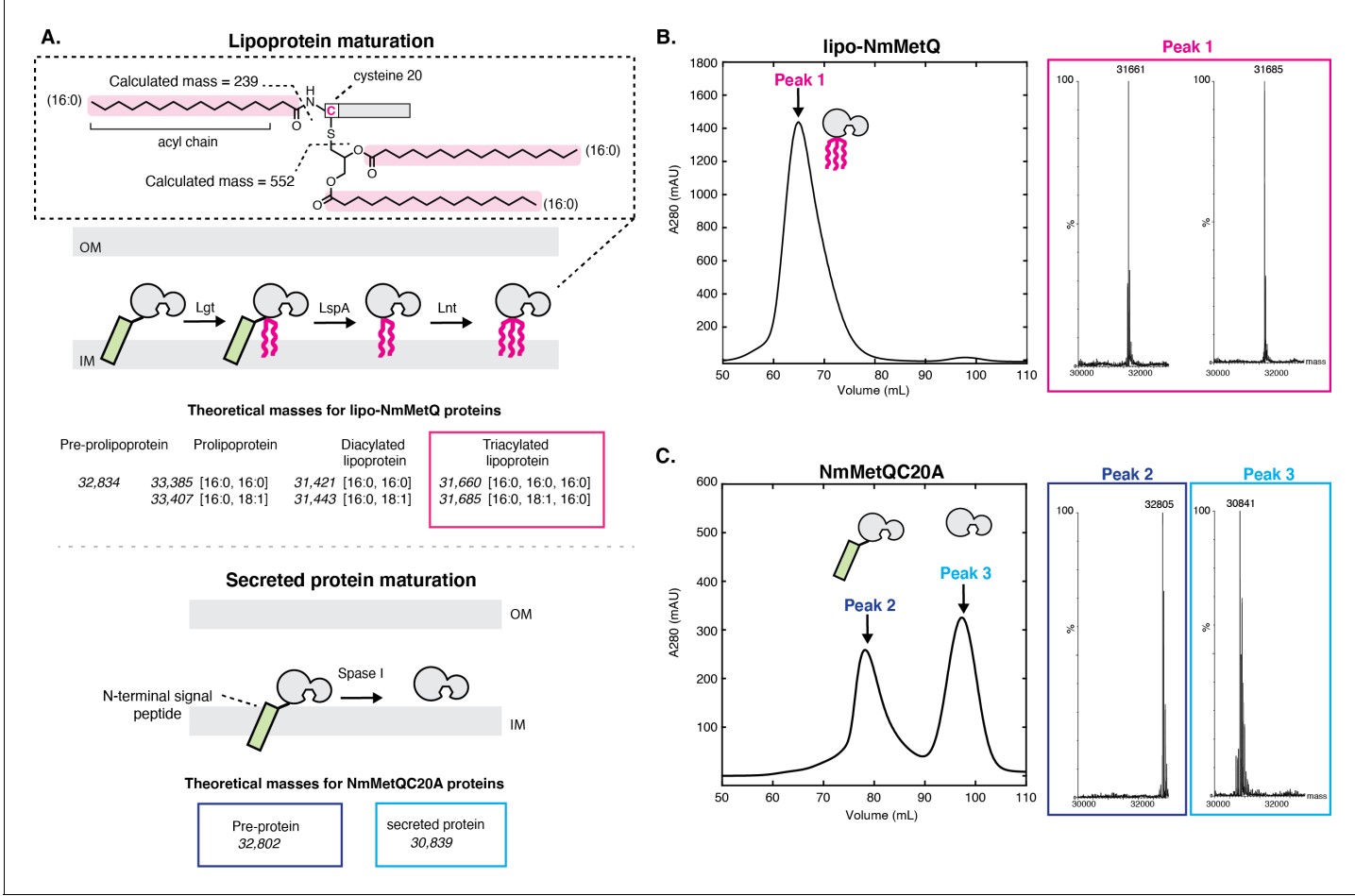

**Figure 1.** Mass spectrometry (MS) analysis of lipo-NmMetQ and NmMetQC20A proteins. (A) (Top) Schematic of lipoprotein maturation pathway. Inset contains a schematic of a lipoprotein with acyl chain composition [16:0,16:0,16:0]. Acyl chains are grouped in a dotted line box and their average masses are calculated. Below the schematic are the theoretical masses for the lipo-NmMetQ proteins (in italics) assuming triacylation occurs via the canonical lipoprotein maturation pathway due to the sequential action of three enzymes (Lgt, LspA, and Lnt). The numbers in the brackets correspond to the total number of carbons and double bonds, respectively, present in the fatty acyl chains of the lipid. (Bottom) Schematic of various NmMetQC20A proteins with example theoretical average masses, shown in italics, assuming cleavage occurs between A19 and A20, possibly by signal peptidase I (SPase I). N-terminal signal peptides are represented by a green rectangle. (B) Characterization of lipo-NmMetQ. Size-exclusion chromatogram and mass spectra of peak 1. The molecular masses of the major species correspond within 1 Da to the predicted mass for two triacylated NmMetQ species, one with acyl chain composition [16:0, 16:0, 16:0] (31,661 Da) and the other with [16:0, 16:0, 18:1] (31,685 Da). (C) Characterization of NmMetQC20A. Size-exclusion chromatogram and mass spectra of the major species from peak 2 and peak 3. The molecular masses of the major species of peak 2 and 3 correspond to the pre-protein NmMetQ (32,802 Da) and secreted NmMetQ (30,839 Da), respectively. These measured masses are within 3 Da of the predicted masses for each species. Assigned NmMetQ species are depicted in cartoon form on the chromatograms.

The online version of this article includes the following figure supplement(s) for figure 1:

**Figure supplement 1.** DLS and SEC-MALS measurements of NmMetQ proteins.

To confirm that lipid attachment site occurs at the N-terminal Cys 20 on NmMetQ, we generated a Cys-to-Ala NmMetQ mutant (NmMetQC20A). We hypothesized that this mutation would prevent lipid attachment and lead to the accumulation of pre-protein NmMetQ, containing an unprocessed N-terminal signal sequence and the C20A mutation. The NmMetQC20A protein was expressed and purified in DDM as previously described. The SEC elution profile reveals two major peaks with distinct elution volumes, 78 ml and 100 mL for peak 1 and 2, respectively (*Figure 1C*). For peak 1, analysis of the fraction containing the highest peak revealed a deconvoluted mass of 32,804, which correlates well with the theoretical intact mass of the pre-protein NmMetQ (32,802 Da). For peak 2, the deconvoluted mass was 30,840, which agrees with the theoretical intact mass of a secreted

NmMetQ protein cleaved between Ala 19 and Ala 20 (30,839 Da) (*Figure 1A*, bottom). The production of the secreted NmMetQ was surprising since we only expected the accumulation of the pre-protein NmMetQ. However, these data suggest that the Cys-to-Ala mutation created a noncanonical cleavage site, possibly allowing Spase I to inefficiently cleave the pre-protein to yield secreted NmMetQ. Together, these data clearly demonstrate that the major species of recombinantly-expressed NmMetQ is heterogeneously triacylated at Cys 20. Mutating Cys 20 to Ala prevents the production of lipoprotein NmMetQ, leading to the formation of pre-protein NmMetQ and secreted NmMetQ. The location of cleavage site, position of lipid attachment, and heterogeneous triacyl chain composition of NmMetQ in this study are consistent with previous studies characterizing other lipoproteins produced in *E. coli* (*Luo et al., 2016*; *Kwok et al., 2011*).

These data also reveal an interesting property of each DDM-solubilized NmMetQ variant: lipo-NmMetQ, pre-protein lipo-NmMetQ, and secreted NmMetQ proteins elute at different volumes despite their similar molecular masses (between 31 and 33 kDa). Specifically, lipo-NmMetQ and pre-protein NmMetQ proteins elute at much higher apparent mass than secreted NmMetQ on a HiLoad 16/600 Superdex 200 (GE healthcare) column (*Figure 1B,C*). To further investigate the properties of the NmMetQ proteins, we used dynamic light scattering (DLS) to measure their hydrodynamic radii ($R_h$) and calculate their theoretical molecular weights assuming a folded globular protein. We found that the $R_h$ values and molecular weight estimates were larger for lipo-NmMetQ ($R_h$ = 7.9 ± 0.2 nm, Mw-R = 430 ± 20 kDa) and pre-protein NmMetQ ($R_h$ 7.7 ± 0.06 nm, Mw-R = 400 ± 7 kDa) than for secreted NmMetQ ($R_h$ 3.0 ± 0.013 nm, Mw-R = 43.0 ± 0.3 kDa) (*Figure 1—figure supplement 1*). These proteins were also analyzed using Size Exclusion Chromatography with Multi-Angle Light Scattering (SEC-MALS). For both lipo-NmMetQ and pre-protein NmMetQ, estimated molar masses were lower with SEC-MALS when compared to DLS, with lipo-NmMetQ measurements of 111 ± 0.3 versus 430 ± 20 kDa and pre-protein NmMetQ measurements of 105 ± 0.3 versus 400 ± 7 kDa, respectively. These data suggest that both lipo-NmMetQ and pre-protein NmMetQ aggregate and that the mass of the aggregate depends on the precise condition of the experiment. However, molar masses for the secreted NmMetQ protein were more similar (26 ± 1 versus 43.0 ± 0.3 kDa), suggesting that unlike lipo-NmMetQ and pre-protein NmMetQ, secreted NmMetQ does not associate with DDM micelles. Based on the size-exclusion chromatograms, DLS, and SEC-MALS data, we propose that both lipo-NmMetQ and pre-protein NmMetQ aggregate with DDM to form protein-DDM micelle-like complexes.

## The ATPase activity of NmMetNI is maximally stimulated in the presence of both lipo-NmMetQ and L-methionine

*Figure 2A* shows that in the presence of 1 μM NmMetNI alone (black trace) and in the presence of 50 μM L-methionine (blue trace), the ATPase activity was low, demonstrating that L-methionine alone is not sufficient to stimulate NmMetNI ATPase activity. However, in the presence of both 1 μM lipo-NmMetQ and 50 μM L-methionine, a marked stimulation of ATPase activity was observed (*Figure 2A*, green trace). To exclude the possibility that the stimulation of ATPase activity is mediated by either the lipid-moiety or the unliganded NmMetQ protein subunit, the experiment was repeated in the absence of L-methionine (NmMetNI and unliganded lipo-NmMetQ only) (*Figure 2A*, magenta trace). Under these conditions the ATPase activity is low, showing that unliganded lipo-NmMetQ is not sufficient to stimulate NmMetNI activity. Given these findings, we conclude that NmMetNI ATPase activity is tightly coupled, requiring both L-methionine and lipo-NmMetQ for maximum stimulation. This result strongly suggests that lipo-NmMetQ plays a role in methionine-mediated NmMetNI ATP hydrolysis.

Next, we characterized the effect of different NmMetQ proteins (lipo-NmMetQ, pre-protein NmMetQ, and secreted NmMetQ) on the ATPase activity of NmMetNI. *Figure 2B* demonstrates that in the presence of 50 μM L-methionine, the NmMetNI ATPase activity increases with increasing concentration of lipo-NmMetQ up to 2 μM, after which the activity starts to plateau (green trace). The same protocol was performed with pre-protein NmMetQ, which contains an N-terminal signal sequence but without the lipid modification. Addition of pre-protein NmMetQ also led to stimulation of ATPase activity, although to a lesser extent than observed for lipo-NmMetQ (orange trace). Addition of secreted NmMetQ, however, had little effect on the ATPase activity (cyan). Together, these data establish that the lipid moiety of lipo-NmMetQ is required for maximal NmMetNI

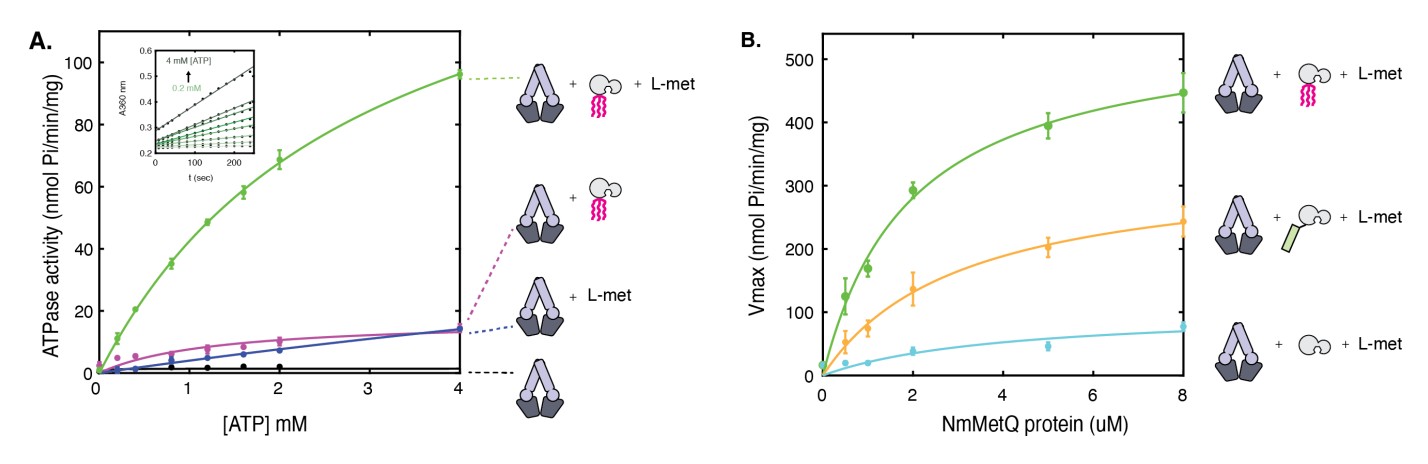

**Figure 2.** ATP hydrolysis of NmMetNI in the presence and absence of L-methionine and NmMetQ proteins. (**A**) ATP hydrolysis was measured in the presence of 1 μM of DDM-solubilized NmMetNI alone (black trace), 50 μM L-methionine (blue trace), 1 μM lipo-NmMetQ (magenta trace) and both 50 μM L-methionine and 1 μM lipo-NmMetQ (green trace). Insert shows representative measurements of absorbance versus time (black dots) and the linear fits (green lines) for NmMetNI ATPase activity in the presence of lipo-NmMetQ and L-methionine at increasing ATP concentrations (0.2, 0.4, 0.8, 1.2, 1.6, 2.0, and 4.0 mM) (**B**) Specific activity of NmMetNI with increasing concentrations of various NmMetQ proteins: lipo-NmMetQ (green trace), pre-protein NmMetQ (orange trace), and secreted NmMetQ (cyan trace) with 50 μM L-methionine. Vmax values were determined by fitting the Michaelis-Menten equation to a plot of ATPase activity versus ATP concentration (0.2, 0.4, 0.8, 1.2, 1.6, 2.0, and 4.0 mM) at different MetQ protein concentrations ( 0.5, 1, 2, 4, 5, 8 μM). N=3 error bars represent standard error of the mean (SEM). These data show the NmMetNI ATPase activity is tightly coupled, requiring both L-methionine and lipo-NmMetQ for maximal NmMetNI ATPase stimulation.

stimulation, although the N-terminal signal sequence of pre-protein NmMetQ could partially mimic its stimulatory effect.

A comparison of NmMetNI's ATPase activity with that of the previously characterized EcMetNI reveals that these transporters have different ligand-dependent ATPase activities. When L-methionine and SBP are absent, NmMetNI has no detectable basal ATP activity. However EcMetNI has a basal ATPase rate of 300 nmol Pi/min/mg (*Kadaba et al., 2008*). These transporters also differ in their response to L-methionine. In the presence of L-methionine, the ATPase activity of EcMetNI decreases due to the binding of L-methionine to the C2 domain, which is responsible for the regulatory phenomenon of transinhibition. For NmMetNI, however, no such effect was detected, as anticipated from the absence of the C2 autoinhibitory domain in NmMetNI.

A comparison of NmMetNI SBP-dependent ATPase stimulation to other ABC importers also reveals some similarities and differences. For NmMetNI, only liganded-SBP maximally stimulated NmMetNI ATP hydrolysis. Maximal stimulation by liganded-SBPs is also a mechanistic feature shared by the ABC importers EcMalFGK₂ (*Davidson et al., 1992*) and EcHisQMP₂ (*Ames et al., 1996*). In contrast, for the ABC importer EcYecSC-FliY (*Sabrialabed et al., 2020*), full stimulation of ATPase can be achieved in both the liganded-SBP and the unliganded-SBP. Although the origin of these differences are unclear, our data show that NmMetNI is tightly coupled and highlight the mechanistic differences between ABC importers.

## N-formyl-L-methionine, L-norleucine, L-ethionine, and L-methionine sulfoximine are potential substrates for the lipo-NmMetQ:NmMetNI system

To identify potential substrates of the NmMetNI-lipoprotein MetQ system, we determined the relative binding affinities of several methionine analogs to NmMetQ. For these measurements, we used Fluorine chemical shift Anisotropy and eXchange for Screening (FAXS) in competition mode, a powerful solution NMR experiment that monitors the displacement of a fluorine-containing reporter molecule by a competing ligand. An important feature of FAXS is that fluorine modification of the competing ligand is not required (*Dalvit et al., 2003*; *Dalvit and Vulpetti, 2019*). As previously discussed (*Gerig, 1994*; *Dalvit and Vulpetti, 2019*), the fluorine nucleus has several properties that are advantageous for NMR: $^{19}F$ is 100% abundant, possesses a spin 1/2 nucleus, and has high

gyromagnetic ratio, which results in high sensitivity (83% of $^1$H). It also has a large chemical shift anisotropy (CSA), allowing higher responsiveness to change in molecular weight, such as those that occur during a protein-ligand binding event. Additionally, since fluorine atoms are not present in most commonly used buffer systems and virtually absent from all naturally occurring biomolecules, background interference in fluorine NMR experiments is minimal.

To optimize the FAXS experiment, we considered several factors. As shown in *Figure 1B*, lipoprotein NmMetQ may multimerize, possibly through an association with the hydrophobic acyl chains, increasing its apparent molecular weight. Because FAXS is sensitive to the apparent molecular weight of the protein, we chose to use a NmMetQ construct lacking its native N-terminal signal sequence and is therefore not modified with lipids (referred to here as NLM-NmMetQ). Trifluoromethyl-methionine was selected as a reporter molecule and the fluorine signal intensity was monitored in the presence of NLM-NmMetQ and several methionine analogs (*Figure 3A*). For these studies, we optimized the concentration of the reporter molecule, NLM-NmMetQ, and the relaxation time (T2) for the NMR measurement. A reporter molecule concentration of 2 mM was chosen here to decrease acquisition time. Additionally, 43 µM NLM-NMMetQ was chosen as a compromise between using less protein and increasing the fraction of reporter bound to the protein. The relaxation time T2 = 120 ms was chosen for its ability to strongly attenuate but not eliminate the reporter signal in the presence of 43 µM NLM-NmMetQ. As previously described (*Dalvit et al., 2003*; *Dalvit and Vulpetti, 2019*), for all experiments, two fluorine spectra (1D and Car-Purcell-Meiboom-Gill (CPMG) filtered) were acquired. The intensity signals of the reporter molecule measured in both spectra and the ratio -ln(CPMG/1D) were calculated. We anticipated that analogs that bind to NLM-NmMetQ would lead to the displacement of the reporter molecule, resulting in a decrease in the -ln(CPMG/1D) ratio.

Our results for the competition binding experiments are shown in *Figure 3C*. The plot shows the signal intensity ratio of the reporter molecule in the presence of each methionine analog. Since displacement of the reporter molecule by the analog correlates to the analog's binding affinity, methionine analogs with higher affinity will be positioned toward the left side of the plot, while lower affinity methionine analogs will appear on the right side. As controls, we measured the -ln(CPMG/1D) ratios with the reporter molecule alone and reporter molecule plus NLM-NmMetQ. As expected, the reporter molecule alone has a low -ln(CPMG/1D) ratio, while the reporter molecule plus NLM-NmMetQ has a high -ln(CPMG/1D) ratio (less free reporter molecule due to NLM-binding).

Next, we carried out the FAXS experiments in the presence of various methionine analogs. We first added L-methionine, a known high-affinity ligand of NmMetQ (Kd 0.2 nM *Nguyen et al., 2019*). As expected for a higher affinity ligand, L-methionine completely displaced the reporter molecule. We then examined two methionine analogs with amino group substitutions: (1) N-formyl-L-methionine, which is used by bacteria to initiate translation and (2) N-acetyl-L-methionine, which is present in bacteria (*Schmidt et al., 2016*) and human brain cells (*Smith et al., 2011*; *Figure 3C*, circles). Addition of these analogs led to the respective complete or near complete displacement of the reporter molecule, indicating that changes to the amino group do not dramatically affect the substrate's ability to bind tightly to NLM-NmMetQ. D-methionine displaced less reporter than L-methionine, consistent with its lower binding affinity (3.5 µM *Nguyen et al., 2019*), while N-acetyl-D methionine failed to displace the reporter molecule. These results suggest that modifications to both the amino group and stereochemistry lead to significantly weaker binding than the singly modified derivative.

Similar to our observations with D-methionine and N-acetyl-D methionine, changes to the carboxyl group resulted in less displacement of the reporter molecule than L-methionine. Specifically, L-methioninol, with the carboxyl group reduced to an alcohol, failed to displace the reporter molecule while L-methionine ethyl ester only partially displaced the reporter molecule (*Figure 3C*, circles).

Lastly, changes to the L-methionine side-chain exhibited varying effects. Methionine analogs with changes to the sulfur atom, including seleno-L-methionine, L-methionine sulfoximine, and L-norleucine were well tolerated, with a greater displacement of the reporter molecule than D-methionine, which has an estimated Kd of 3.5 µM (*Nguyen et al., 2019*). However, L-ornithine failed to displace the reporter molecule, suggesting that binding of ligands with a charged amino group is energetically unfavorable. Side-chain length also plays a role in methionine analog affinity to NLM-NmMetQ. Increasing the side-chain length by an addition of a methylene group (L-ethionine) was better

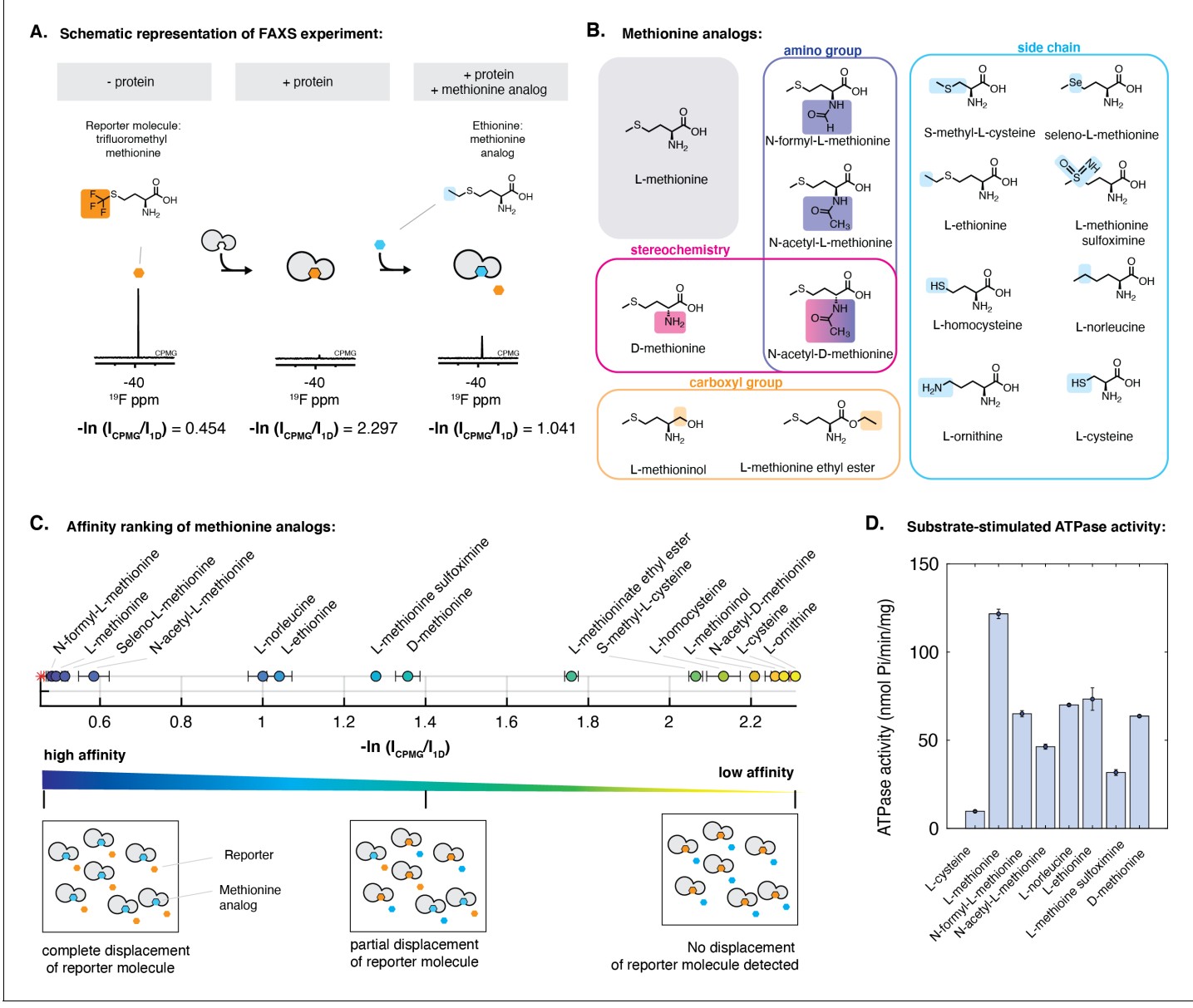

**Figure 3.** Characterization of the interaction of methionine analogs with NmMetQ using FAXS and ATPase experiments. (A) Schematic diagram of the FAXS experiment. The intensity of the fluorine signal decreases in the presence of NLM-NmMetQ. Addition of the methionine analog causes the fluorine signal intensity of the reporter molecule to increase due to its displacement from NLM-NmMetQ. (B) Chemical structures of the methionine analogs used in this study. (C) (Top) Ordering of methionine analogs by their binding affinity to NLM-NmMetQ. (Bottom) Schematic representation of FAXS experiment depicted in bulk solution. Methionine analogs with higher affinity are positioned toward the left side of the plot, while lower affinity methionine analogs are positioned toward the right. (D) ATPase activity of NmMetNI at 2 mM ATP in the presence of lipo-NmMetQ and methionine analogs at 1:8:50 molar ratio, respectively. N=3 error bars represent SEM.

The online version of this article includes the following source data for figure 3:

**Source data 1.** The measured -ln(Icpmg/I1D) values: NMR.xlsx.

tolerated than decreasing the length by one carbon (S-methyl-L-cysteine). Shorter thiol derivatives (L-cysteine and L-homocysteine) were ineffective at displacing the reporter molecule. Together, our data establish that NLM-NmMetQ can accommodate variability in the binding of methionine analogs, including modifications to the amino group, D-stereochemistry, and changes in the side-chain (to a limited extent), while exhibiting little tolerance for variations in the carboxyl group.

To determine whether methionine analogs could serve as potential substrates for the lipo-NmMetQ:NmMetNI system, we measured NmMetNI ATPase activity in presence of lipo-NmMetQ and several methionine analogs. For these assays, we chose methionine analogs identified by FAXS to bind NLM-NmMetQ with an affinity similar or higher than D-methionine, a known substrate for *E. coli* NmMetNI. Since substrate-stimulated ATPase activity is a hallmark of ABC transporters (*Bishop et al., 1989*; *Mimmack et al., 1989*), we expected methionine analogs that are substrates for this system would stimulate NmMetNI ATPase activity. *Figure 3D* shows the results for the methionine analog stimulation of NmMetNI ATPase activity. As a negative control, we tested L-cysteine, where, as expected, no substrate-stimulated ATPase stimulation was detected. Our data show that the following methionine analogs led to substrate-stimulated ATPase activity: N-acetyl-L-methionine, N-formyl-L-methionine, L-norleucine, L-ethionine, and L-methionine sulfoximine. However, no correlation was seen between affinity to NLM-NmMetQ and NmMetNI stimulation. This data suggest that binding to NmMetQ is necessary to initiate transport; however, this step alone does not determine the magnitude of NmMetNI ATPase stimulation. Taken together, our FAXS and ATPase experiments suggest that N-formyl-L-methionine, L-norleucine, L-ethionine, and L-methionine sulfoximine are potential substrates for the *N. meningitidis* lipo-NmMetQ:NmMetNI system.

## Structures of *N. meningitidis* MetNI in the inward-facing conformation and *N. meningitidis* lipo-NmMetQ:NmMetNI complex in the outward-facing conformation

To gain insight into the role of lipo-NmMetQ in the NmMetNI transport cycle, we determined structures of NmMetNI in different conformational states by single-particle cryo-EM. By varying the nucleotide analog and concentration, multiple conditions were screened to identify ones that promoted different conformational states. With 1 mM AMPPNP (below the Km), the structural analysis of an equimolar mixture of lipo-NmMetQ and NmMetNI revealed that NmMetNI was captured in the inward-facing conformation at 3.3 Å resolution (*Figure 4A*); no densities for either AMPPNP and lipo-NmMetQ were observed. For this data set, the two dimensional class averages showed clear structural features, suggesting a high level of conformational homogeneity (*Figure 4—figure supplement 1*). The overall architecture of NmMetNI is similar to previously determined structures of EcMetNI, comprising two copies of each TMD and NBD, encoded by *MetI* and *MetN*, respectively (*Kadaba et al., 2008*; *Johnson et al., 2012*). Each MetI subunit contains five transmembrane helices per monomer for a total of ten transmembrane helices per transporter (*Figure 4B*).

A comparison between NmMetNI and EcMetNI reveals similar subunit folds, with the root mean square deviation (RMSD) of 2.4 Å over 843 Cα atoms. As predicted from the primary sequence, the NmMetN subunits lack the C2 autoinhibitory domain. As a result, the interfaces of NmMetNI and EcMetNI are distinct. In the inward-facing conformation of NmMetNI, the NBDs interact directly. In contrast, in EcMetNI, the inward-facing conformation forms an interface through the C2 autoinhibitory domains, with a slight separation between the NBDs (*Figure 4—figure supplement 2A*). A similar increase in NBD:NBD distance, defined as the average distance between Cα of glycines in the P loop and signature motifs, is observed in the previously determined molybdate ABC transporter structures, *Methanosarcina acetivorans* ModBC (MaModBC) and *Archaeoglobus fulgidus* ModBC (AfModBC) (*Hollenstein et al., 2007*; *Gerber et al., 2008*; *Figure 4—figure supplement 2B*). To date, these are the only other reported pair of homologous structures, one with an autoinhibitory domain and one without. For AfModBC, which lacks the autoinhibitory domain, the NBD:NBD distances are ˜17 Å and 21 Å for each AfModBC in the asymmetric unit. For MaModBC, which does have an autoinhibitory domain, this distance increases to ˜27 Å. A comparison of these structures suggests that type I ABC importers share a common quaternary arrangement in the inward-facing conformation such that the presence of a regulatory domain increases the separation of the NBD:NBD distance in comparison to the homologous structure without a regulatory domain *Figure 4—figure supplement 2D*.

Reasoning that increasing the nucleotide concentration to above the Km for MgATP would promote complex formation, we mixed equimolar lipo-NmMetQ and NmMetQ in the presence of 5 mM ATP. Under these conditions, we were able to determine the single-particle cryo-EM structure of the complex to 6.4 Å resolution (*Figure 4*). Despite extensive efforts, we were unable to improve the resolution of this complex. The structure was modeled by rigid-body refinement of both NmMetNI in the inward-facing conformation (traced from the 3.3 Å resolution reconstruction) and the

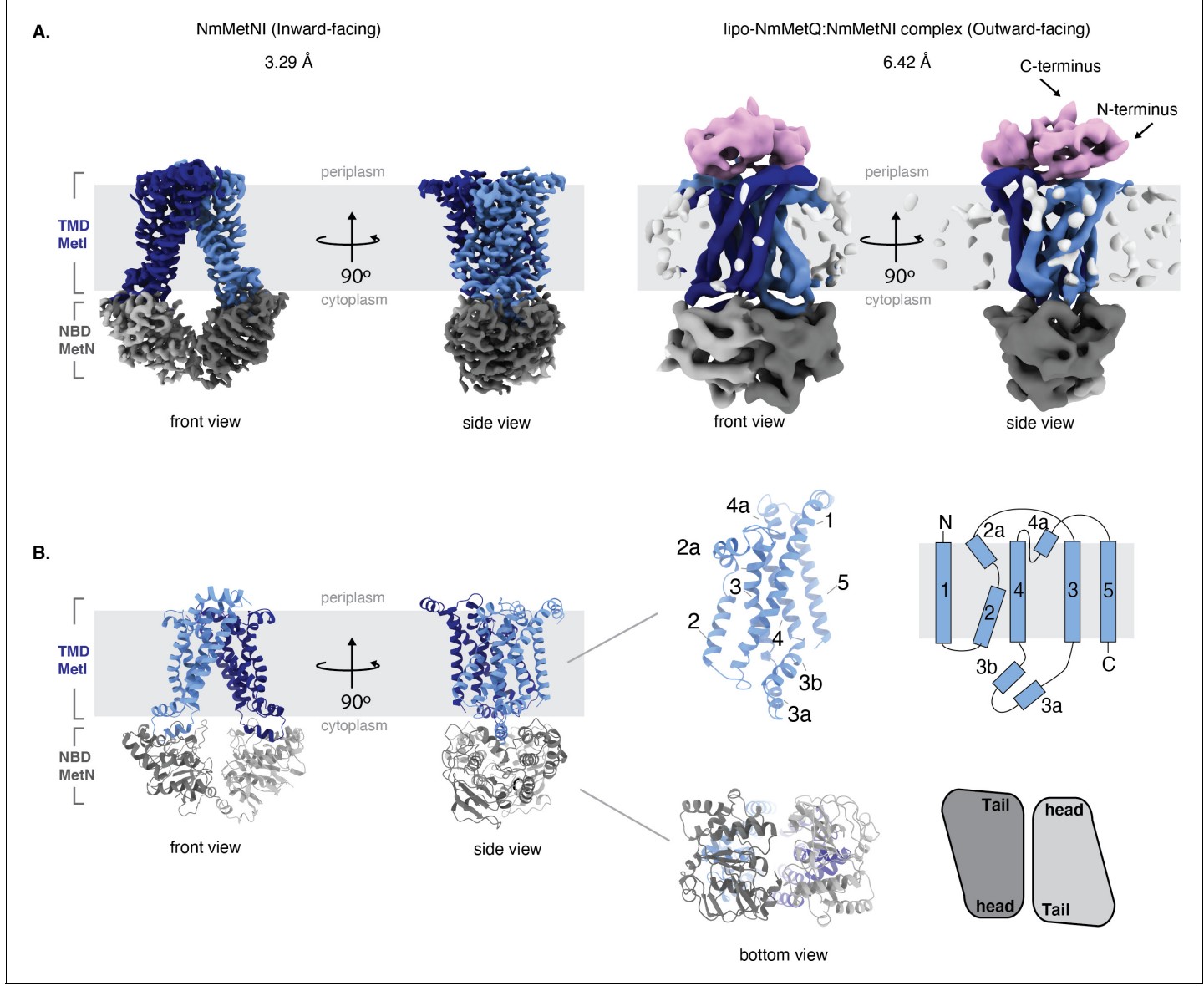

**Figure 4.** Architecture of NmMetNI and lipo-NmMetQ:NmMetNI complex. (**A**) The 3.3 Å resolution cryo-EM map and NmMetNI in the inward-facing conformation in two views. (**B**) Transmembrane localization of NmMetI, showing NmMetI contains five transmembrane helices per monomer (**C**) The 6.4 Å resolution cryo-EM map and model of NmMetNI in complex with lipo-NmMetQ in the presence of ATP. NmMetNI is in the outward-facing conformation. NmMetI is shown in light/dark blue, NmMetN in light/dark grey and lipo-NmMetQ in light pink. The membrane is represented by a gray box.

The online version of this article includes the following figure supplement(s) for figure 4:

**Figure supplement 1.** Cryo-EM map generation and data processing refinement of NmMetNI in the inward-facing conformation.

**Figure supplement 2.** Comparison of type I ABC transporters.

**Figure supplement 3.** Cryo-EM map generation and data processing refinement of lipo-NmMetQ:NmMetNI complex in the outward-facing conformation.

previously determined soluble NmMetQ structure in the substrate-free conformation (PDB:6CVA). Our model shows lipo-NmMetQ docked onto the NmMetI subunits and the NmMetN subunits in a closed dimer state. No clear density was seen for the lipid moiety of lipo-NmMetQ or ATP (*Figure 4*). Of note, Liu et. al. were also not able to observe the lipid moiety of lipo-SBP in complex with an ABC transporter, despite these cryo-EM structures resolving at higher resolutions (3.30, 3.44, 3.78 Å) *Liu et al., 2020*. A comparison between NmMetNI and EcMetNI in the outward-facing

conformation in complex with their respective MetQ proteins reveals they have similar conformations, with RMSD of 2.2 Å over 1048 Cα atoms (*Figure 4—figure supplement 2C*). In contrast to the inward-facing conformation, the NBD:NBD arrangement is similar for both EcMetNI and NmMetNI.

## Lipo-MetQ proteins may be present in a variety of other Gram-negative bacteria

We used a bioinformatics approach to determine if other Gram-negative bacteria could have lipid-modified MetQ proteins. For the analysis, we chose predicted MetQ protein sequences from the InterPro family IPR004872 (of which NmMetQ is a member), restricting the search to Proteobacteria, Taxonomy ID 1224, and 90% identity. The amino acid sequence of the MetQ proteins were then analyzed using SignalP 5.0. *Figure 5* summarizes the results. Our data revealed that lipid-modified MetQ proteins may be present in all classes of Proteobacteria (Alpha, Beta, Gamma, Delta, and Epsilon), with varying number of lipid-modified MetQ proteins in each Order (magenta vs white). These results suggest that lipid modification of MetQ proteins are not restricted to *N. meningitidis*(this work) and *E. coli* (*Tokuda et al., 2007*; *Carlson et al., 2019*) and are likely present in a wide variety of Gram-negative bacteria.

## Discussion

NmMetQ has been previously identified as an OM surface-exposed candidate meningococcal vaccine antigen (*Pizza et al., 2000*), possibly playing a role in bacterial adhesion to human brain endothelial cells (*Kánová et al., 2018*). However, the presence of NmMetQ at the OM challenges the prevailing view that SBPs reside in the periplasm, freely diffusing between the IM and OM (*Thomas and Tampé, 2020*). To better understand whether NmMetQ has lost its ABC transporter-dependent function at the IM and how NmMetQ remains at the surface of the bacterium, we used multiple biophysical techniques to characterize the structure and function of NmMetQ and

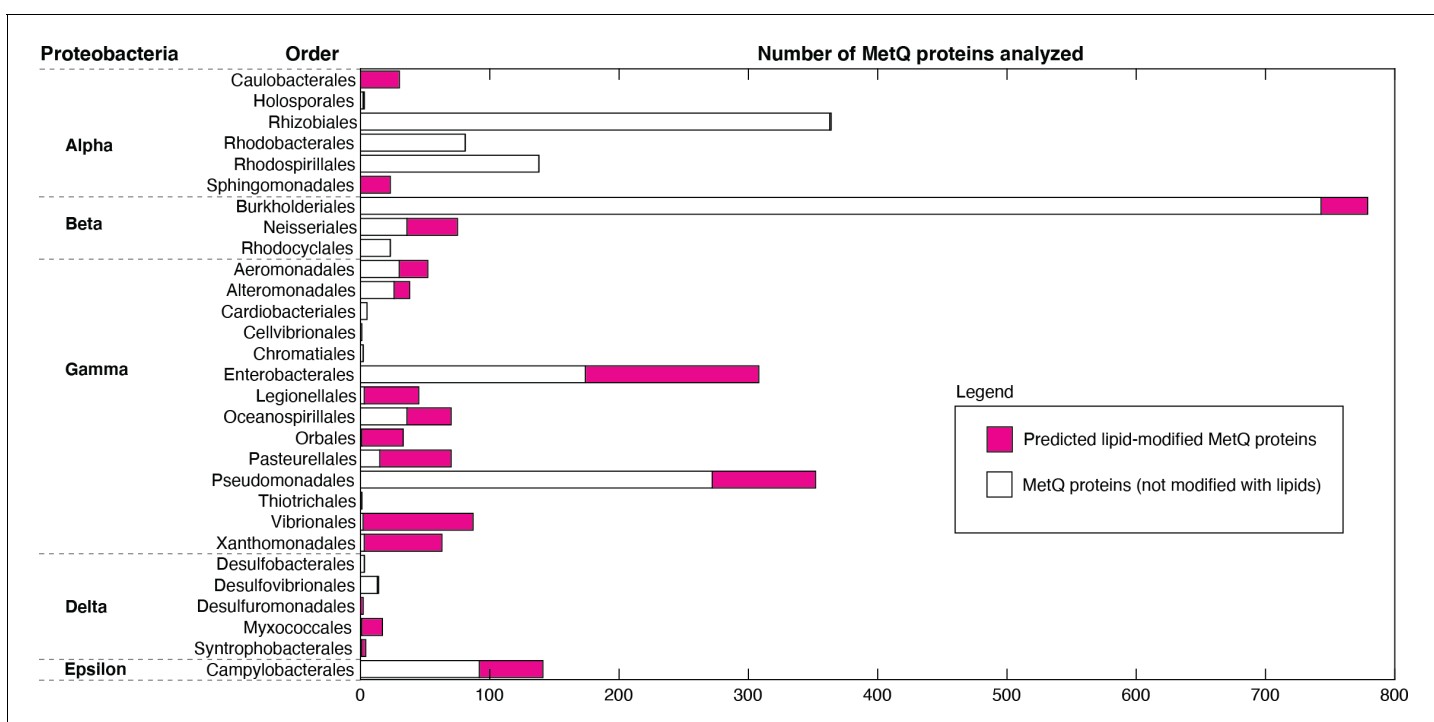

**Figure 5.** Distribution of lipid-modified MetQ proteins in different classes of Proteobacteria, a major phylum of Gram-negative bacteria. Plot of the number of MetQ proteins analyzed in each Order, grouped by Proteobacteria. Predicted lipid-modified and secreted MetQ proteins are shown in magenta and white, respectively.

The online version of this article includes the following source data for figure 5:

**Source data 1.** Distribution of lipid-modified MetQ proteins: lipoproteins.xlsx.

NmMetNI. Here, we show that NmMetQ is a lipoprotein that binds and stimulates the ATPase activity of NmMetNI.

Based on our data, we propose a model for NmMetQ localization that reconciles previous studies identifying NmMetQ as a surface-exposed candidate antigen and our study characterizing NmMetQ as a cognate SBP to NmMetNI. In our model, NmMetQ is a lipoprotein with dual function and localization (*Figure 6*). At the IM, lipo-NmMetQ plays a role in nutrient acquisition. Lipo-NmMetQ is then transported to the OM, possibly through the localization of the lipoprotein system (Lol) (*Zückert, 2014*), and then flipped to the surface of the cell via Slam (a protein involved in lipoprotein surface exposure in *N. meningitidis*) (*Hooda et al., 2016*). Lipo-NmMetQ then anchors to the OM cell-surface via its lipid moiety, playing a role in adhesion. The lipid modifications are central to our model, helping to explain how NmMetQ remains at the surface of the bacterium.

Our identification of NmMetQ as a lipoprotein is predicated on our ability to express and purify lipo-NmMetQ and its processing variants. We recognize that a key assumption in our study is that the *E. coli* and *N. meningitidis* lipoprotein maturation machineries process the N-terminal signal sequences of lipoproteins in a similar manner. Since previous studies have successfully expressed in *E. coli* lipoproteins with their native signal sequences from other Gram-negative bacteria (*Parra et al., 2010*; *Hooda et al., 2016*), including two lipoproteins from *N. meningitidis* (*Fantappiè et al., 2017*), we reasoned that these biochemical pathways are sufficiently similar between *E. coli* and *N. meningitidis* to justify this assumption.

Our ability to express and purify lipo-NmMetQ, pre-protein NmMetQ, and secreted NmMetQ allowed us to carry out in vitro studies investigating whether NmMetQ can function as an SBP for NmMetNI. Functional assays showed that both lipo-NmMetQ and L-methionine are required for maximal ATPase stimulation of NmMetNI. NmMetNI can also be stimulated, although to a lesser extent, by pre-protein NmMetQ/L-methionine, and lipo-NmMetQ with methionine analogs. Binding of lipo-NmMetQ to NmMetNI was also investigated by determining the cryo-EM structures of NmMetNI in the presence and absence of lipo-NmMetQ. Our structures show lipo-NmMetQ binds to the TMDs of NmMetNI, similar to what is observed with the well-characterized *E. coli* ABC

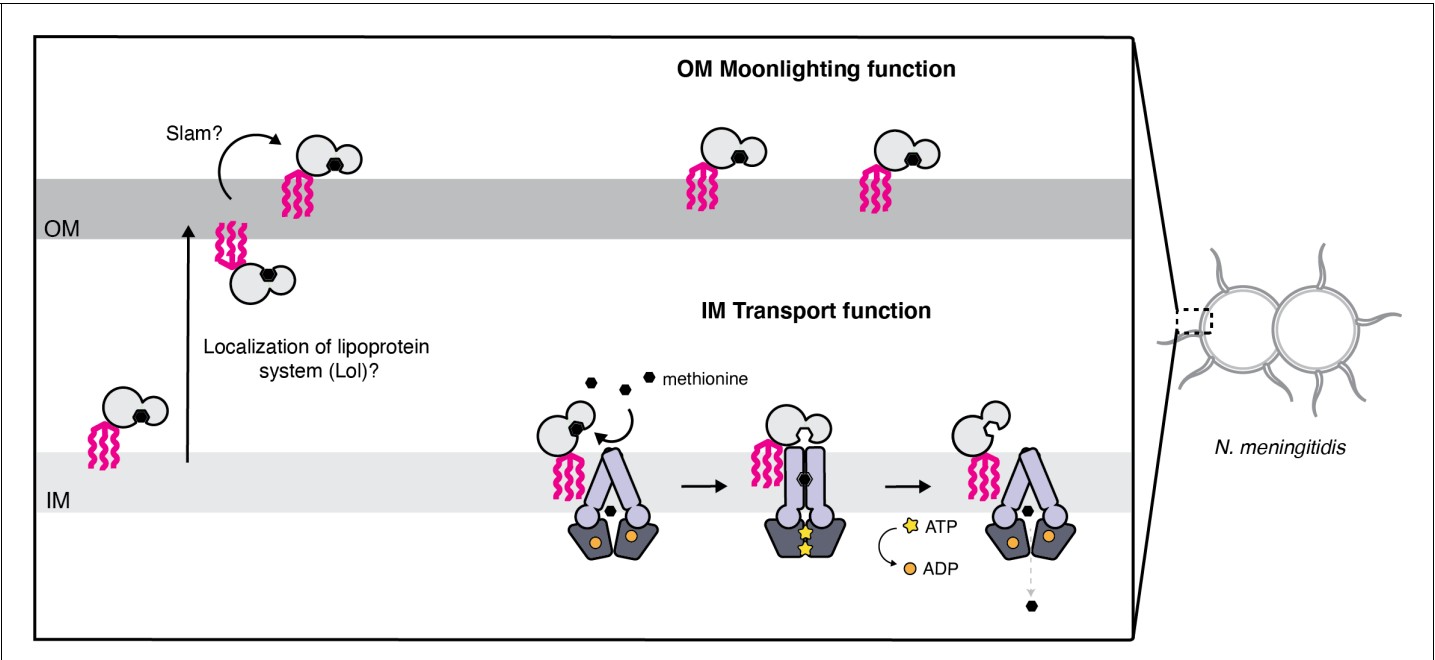

**Figure 6.** Proposed model for the cellular distribution of the *N.meningitidis* ABC methionine transporter proteins. Lipo-NmMetQ tethers to membranes via a lipid anchor and has dual function and localization, playing a role in NmMetNI-mediated transport at the inner-membrane in addition to moonlighting on the bacterial surface. The lipid modifications are central to the model, helping to explain how NmMetQ remains at the surface of the bacterium.

methionine transporter system, EcMetQ:EcMetNI (*Nguyen et al., 2018*). Together, our data suggests that lipo-NmMetQ plays a role in NmMetNI-mediated nutrient acquisition.

The dual functionality of SBPs may help explain why the intracellular concentrations of SBPs are typically 20x that of their cognate ABC transporters, depending on growth conditions (*Schmidt et al., 2016*). Of particular note, under many of the tested growth conditions, MetQ was the most abundant SBP in *E. coli*, present at up to nearly 30,000 copies per cell; for comparison MetNI was present at 1000 copies per cell. A tempting interpretation of this observation is that larger number of SBPs increases the efficiency of nutrient uptake. Given that methionine is a scarce amino acid in human nasopharynx (*Krismer et al., 2014*), where *N. meningitidis* primarily colonizes (*Stephens et al., 2007*), and one of the most expensive amino acids to synthesize in terms of ATP requirement (as measured in *E. coli*) (*Kaleta et al., 2013*), having multiple copies of NmMetQ may enable *N. meningitidis* to more efficiently capture methionine from the nutrient-limited environment.

However, our study raises the possibility that higher SBP concentrations may also reflect SBP participation in ABC transporter-independent functions, including moonlighting functions at the surface of the cell. As a consequence, the stoichiometry of SBPs to cognate ABC transporters measured in Schmidt et al. may be misleading if SBPs are distributed between multiple locations, in addition to the periplasmic space (*Schmidt et al., 2016*). Our study also calls for caution in interpreting SBP gene knock-out experiments, since the deletion of SBPs genes may lead to phenotypes associated with the loss of either or both ABC transporter-dependent and -independent SBP functions.

While previous studies have shown that many SBPs of Gram-negative are soluble (*Heppel, 1969*), our findings suggest that at least some SBPs may be modified with lipids. Since lipid modification may enable SBPs to localize to the surface in Gram-negative bacteria, we believe that future efforts should be made to experimentally determine which SBPs have lipid modifications, dual localization, and ABC transporter-independent functions. Studies aimed at determining the rules that govern protein surface-exposure will not only increase our understanding of bacterial physiology but will also help in the rational design of vaccines based on surface-exposed protein antigens.

## Materials and methods

### Cloning, expression, and purification of *N. meningitidis* proteins

The protein encoding genes of MetQ and MetNI were obtained from *N. meningitidis* virulent strain MC58, GeneBank accession number AE002098. To produce MetNI, the DNA sequences encoding both MetN and MetI were inserted into a single modified pET vector, each under the control of a separate T7 promoter. To aid expression and purification, a decahistidine plus enterokinase site MGHHHHHHHHHHHSSGHIDDDKH sequence was added to the N-terminus of MetN, while MetI contained no additional residues. A similar strategy was used to produce other ABC transporters (*Locher et al., 2002*; *Pinkett et al., 2007*). To produce lipo-NmMetQ, the DNA sequence encoding the NmMetQ with the native signal sequence and a C-terminal decahistidine tag was added to a single modified pET vector. This construct served as a template to generate the C20A mutant, which was created using PCR site-directed mutagenesis. NLM-NmMetQ was created as previously described (*Nguyen et al., 2019*).

All proteins were expressed in *E. coli* BL21 (DE3) gold cells (Agilent Technologies) using autoinduction media (*Studier, 2005*) by growing cells for 48 hr at 22 °C. Cells were harvested by centrifugation and stored at −80 °C. To purify lipo-NmMetQ, NmMetQC20A proteins (pre-protein and secreted NmMetQ) and the transporter NmMetNI, 10 grams of frozen cell paste were thawed and homogenized in 100 mL of ice cold lysis buffer 25 mM Tris, pH 7.5, 100 mM NaCl, 40 mg of lysozyme, 4 mg DNase, and 1 Complete Protease Inhibitor Cocktail Tablet (Roche Diagnostics GmbH). Cells were lysed by the addition of 1% v/w n-dodecyl-β-D-maltopyranoside (DDM, Anatrace) and by stirring the homogenate for 3 hr at 4 °C. Cell debris was removed by 45,000 rpm centrifugation for 45 min. Proteins were purified using a 5 mL HisTrap HP column (GE healthcare) followed by gel filtration (HiLoad 16/600 Superdex 200 GE healthcare), pre-equilibrated with 25 mM Tris, pH 7.5, 100 mM NaCl, and 0.05% DDM. Proteins were stored at −80 °C until thawed.

## Dynamic light scattering

DLS measurements were performed using a DynaPro NanoStar instrument (Wyatt Technology) using the manufacturer's suggested settings. A disposable UVette cuvette (Eppendorf) was used to contain the samples. Each sample was analyzed in triplicate to yield an average and standard deviation. Dynamics 7.1.7 software was used to analyze the data (Wyatt Technology).

## SEC-MALS

Proteins were loaded onto a Superdex Increase 10/300 GL column (GE healthcare) pre-equilibrated with 25 mM Tris, pH 7.5, 100 mM NaCl, and 0.05% DDM and passed through a Wyatt DAWN multiangle light scattering detector, equipped with WyattQELS dynamic light scatting (DLS) module, coupled to a Wyatt Optilab refractive index detector (Wyatt Technology). Astra 7.3.2.19 software was used to analyze the data (Wyatt Technology). The same lipo-NmMetQ, pre-protein NmMetQ, and secreted NmMetQ protein stocks were thawed and used for both DLS and SEC-MALS measurements.

## Single-particle Cryo-EM

Sample for the structure of NmMetNI in the inward-facing conformation was prepared by mixing equal volumes of lipo-NmMetQ and NmMetNI (1.3 and 4.6 mg/ml, respectively) and 100 mM Adenylyl-imidodiphosphate (AMPPNP) to a final concentration of 1 mM. The mixture was then concentrated six-fold using a MilliporeSigma Amicon Ultra Centrifugal Filter Unit with a cellulose membrane of 100 kDa (Thermo Fisher Scientific). Sample for the structure of the lipo-NmMetQ:NmMetNI complex in the outward-facing conformation was prepared by mixing equal volumes of lipo-NmMetQ and NmMetNI and addition of 100 mM ATP to a final concentration of 5 mM.

UltrAufoil 1.2/1.3, 300 mesh grids (Electron Microscopy Sciences) were glow-discharged for 60 s at 15 mA using a PELCO easiGLOW (Ted Pella). Samples were then incubated at 37 °C for 5 min and then applied to the grids (3 μL), blotted with Whatman No.1 filter paper for 4 s with a blot force of 0 at 22 °C and 100% humidity and plunge-frozen into liquid ethane using a Mark IV Vitrobot (Thermo Fisher). The grids were then stored in liquid nitrogen until further use.

Data collection was performed in a 300-KeV Titan Krios transmission electron microscope (Thermo Fisher Scientific) at the cryo-EM facility at Caltech in Pasadena, California. Movies were collected using SerialEM v.3.7 automated data collection software (*Mastronarde, 2005*) with a beam-image shift over a three-by-three pattern of 1.2 μm holes with three exposures per hole in super-resolution mode (pixel size of 0.428 Å px$^{-1}$) on a K3 camera (Gatan).

## Image processing

Data collection parameters are summarized in *Table 1*. The data-processing workflow described below was performed for all data sets using cryoSPARC v.2.15 (*Punjani et al., 2017*). Cryo-EM movies were patch motion corrected for beam-induced motion including dose weighting with cryoSPARC after binning super-resolution movies. The non-dose-weighted images were used to estimate CTF parameters using Patch CTF job in cryoSPARC. Micrographs containing either ice or poor CTF fit resolution estimations were discarded. A subset of images was randomly selected and used for reference-free particle picking using Blob picker. Particles were subjected to multiple rounds of 2D classification, and two classes (top and side) were used as templates for particle picking on the full set of images. The subsequent processing steps were different for the two data sets.

For the data set acquired for NmMetNI in the inward-facing conformation, initial particle stacks were extracted, downsampled four times, and then subjected to 2D classification. Classes that were interpreted as junk were discarded. The selected particles were then used to generate *ab initio* volumes. Two volumes, interpreted as NmMetNI and a junk/noise class, were selected for heterogeneous refinement. Particles assigned to the NmMetNI class were processed further by repeating the same strategy using particles downsampled twice, and then again with no downsampled particles. The final resulting particle stack was then non-uniformly refined (*Figure 4—figure supplement 1*).

For the data set acquired for the lipo-NmMetQ:NmMetNI complex in the outward-facing conformation, initial particle stacks were extracted, downsampled ten times and subjected to 2D classification. Classes that were interpreted as junk were discarded. 2D classification was then repeated with particles downsampled by four, and then again with no downsampled particles. The selected

**Table 1.** Cryo-EM data collection and refinement statistics.

| | Inward-facing conformation of the MetNI methionine ABC transporter | Outward-facing conformation of the MetNI methionine ABC transporter in complex with lipo-MetQ |
|---|---|---|
| PDB | 7MC0 | 7MBZ |
| EMD | EMD-23752 | EMD-23751 |
| Data collection conditions | | |
| Microscope | Titan Krios | Titan Krios |
| Camera | Gatan K3 Summit | Gatan K3 Summit |
| Magnification | 105,000x | 105,000x |
| Voltage (kV) | 300 | 300 |
| Recording mode | counting | counting |
| Frames/Movies | 40 | 40 |
| Total Electron dose (e-/Å$^2$) | 60 | 60 |
| Defocus range (μm) | 1.0 – 2.8 | 1.0 – 2.8 |
| Pixel size (Å) | 0.856 | 0.856 |
| Micrographs collected | 4709 | 6183 |
| Micrographs used | 3968 | 5494 |
| Total extracted particles | 1,684,719 | 2,874,862 |
| Refined particles | 322,171 | 58,434 |
| Symmetry imposed | C1 | C1 |
| Nominal Map Resolution (Å) | | |
| FSC 0.143 (unmasked/masked) | 3.4/3.3 | 6.4/6.4 |
| Refinement and Validation | | |
| Initial model used | 3TUJ | |
| Number of atoms | | |
| Protein | 7092 | 8987 |
| Ligand | 0 | 0 |
| MapCC (mask/box) | 0.80/0.65 | 0.75/0.69 |
| Map sharpening B-factor | 91.3 | 496 |
| R.m.s. deviations | | |
| Bond lengths (Å) | 0.012 | 0.012 |
| Bond angles (°) | 1.62 | 1.92 |
| MolProbity score | 1.76 | 1.73 |
| Clashscore (all atom) | 7.56 | 6.77 |
| Rotamer outliers (%) | 1.19 | 1.04 |
| Ramachandran plot | | |
| Favored (%) | 95.77 | 95.09 |
| Allowed (%) | 3.90 | 4.91 |
| Outliers (%) | 0.33 | 0 |

particles were then used to generate *ab initio* volumes. Two volumes, interpreted as lipo-NmMetQ: NmMetNI complex and junk/noise classes were selected for heterogeneous refinement. Particles assigned to the lipo-NmMetQ:NmMetNI complex class were subjected to another round of *ab initio*, followed by heterogeneous refinement. The final resulting particle stack was then non-uniformly refined (*Figure 4—figure supplement 3*).

To build the atomic model of NmMetNI in the inward-facing structure, the structure of EcMetNI (PDB: 3TUJ) lacking the C2 domain was used as template for model building. The model was built by rigid-body docking, homology modeling, and manually building into the 3.3 Å resolution cryo-EM density in Coot v0.9.1 (*Emsley et al., 2010*) and refined using ISOLDE (*Croll, 2018*). The model of the lipo-NmMetQ:NmMetNI complex in the outward-facing conformation was built by rigid-body refinement, using the following templates for model building: (1) NmMetNI in the inward-facing conformation (traced from the 3.3 Å resolution reconstruction) and (2) the previously determined soluble NmMetQ structure in the substrate-free conformation (PDB:6CVA) *Nguyen et al., 2019*. The model was built by rigid-body docking in Coot, followed by refinement in ISOLDE using adaptive distance restraints.

Intersubunit distances between ATP-binding domains were defined by the positions of Cα of glycine residues of the P loop and signature motifs like previously described (*Kadaba et al., 2008*). Specifically, Gly44/Gly144 and Gly43/Gly143 for NmMetNI and EcMetNI (3TUJ), respectively and Gly36/Gly129 and Gly38/Gly130 for AfModBC (2ONK) and MaModBC (3D31), respectively. For each transporter, two intersubunit distances were measured and averaged using UCSF Chimera version 1.1 (*Pettersen et al., 2021*; *Goddard et al., 2018*).

RMSD measurements were carried out with Coot v0.9.1 using SSM Superposition using default settings (*Krissinel and Henrick, 2004*). All images of models and densities were prepared using UCSF Chimera version 1.1.

## MS analysis

The molecular masses of the proteins were determined by Ultra-Performance Liquid Chromatography-Mass Spectrometry (UPLC-MS). The UPLC-MS consisted of a Waters Acquity Chromatography platform and a Waters LCT Premier XE mass spectrometer. The chromatography separations used a solvent system of 0.1% formic in water (solvent A) and 0.1% formic acid in acetonitrile (solvent B), with a 10 min solvent program that reached 95% B at 7 min. UPLC solvent flow was 0.4 mL/min from 0 to 1 min for desalting and was subsequently reduced to 0.22 mL/min. Samples dissolved in 25 mM Tris HCl pH 7.5, 100 mM NaCl, and 0.05% DDM were injected onto a Waters BEH C4 1.7 µ 300 Å 50 mm long 2.0 mm internal diameter column connected directly to the mass spectrometer. Electrospray ionization was used in positive ion mode. The mass spectrometer was operated in the V Mode.

## ATPase experiments

Activity assays were performed in an Infinite 200 microplate reader (Tecan) at 37 °C using the EnzChek phosphate assay kit (ThermoFischer Scientific) to measure the amount of inorganic phosphate. Each 100 µL reaction contained 5 µM NmMetNI, 20 mM Tris-HCl pH 7.5, 100 mM NaCl, 5 mM β-mercaptoethanol, 200 µM 2-amino-6-mercapto-7-methylpurine riboside substrate, 0.1 units of purine nucleoside phosphorylase, and 0.05% DDM. NmMetQ proteins and L-methionine were present as indicated in the figure captions. Samples were incubated for 15 min at 37 °C and the reactions were then initiated by an automatic injection of $MgCl_2$ to a final concentration of 5 mM. Initial rates were determined using Matlab software by calculating the linear portion of the change in absorbance at 360 nm as a function of time following the injection of $MgCl_2$. ATP concentrations of 0.2, 0.4, 0.8, 1.2, 1.6, 2, and 4 mM were used for all experiments.

## NMR

$^{19}$F-NMR spectroscopy All NMR spectra were recorded at 300 K with a Bruker Ascend 400 NMR spectrometer equipped with multinuclear iProbe ($^1$H/$^{19}$F, $^{31}$P-$^{109}$Ag) and a 24 position sample changer. CPMG relaxation dispersion $^{19}$F spectra were recorded with a $T_2$ of 1 ms before the acquisition period and 120 ms between the train of 180° pulses. Free induction decay (FID) signals were backward linear predicted to 11 points and apodized with a 1.5 Hz Lorentzian filter. The spectra

were analyzed with MestReNova v12.0.2 (Mestrelab Research), and intensity values were performed using the Line Fitting tool. Trifluoromethyl methionine was synthesized and purchased from Peptech (Bedford, MA). For the competition experiments, each sample contained 43 µM *N. meningitidis* MetQ, 2 mM trifluoromethyl methionine (reporter molecule), and 43 µM methionine analog (competing molecule).

### Bioinformatics

Protein sequences were obtained through the UniProtKB database using the following search terms: Proteobacteria (taxonomy ID 1224), InterPro family IPR004872 (which NmMetQ UniProt ID Q7DD63 is a member), and identity 90%, which groups sequences with > 90% identity and 80% sequence length. SignalP 5.0 was used separately to analyze the N-terminal protein sequences and predict the location of the signal sequence cleavage sites. Sequence alignment data was generated by the EFI Enzyme Similarity Tool (https://efi.igb.illinois.edu/efi-est/) using Option C with FASTA header reading (*Gerlt et al., 2015*). A SSN network was then created using an alignment score corresponding to approximately 60% sequence identity and filtering for sequences between 240 and 330 residues in length. Cytoscape v3.8.0 (*Smoot et al., 2011*) was used for visualizing lipo-MetQ trends and obtaining taxonomy information. The table was exported and graphed in Matlab (MathWorks).

## Acknowledgements

We thank Jacob Parres-Gold and Dr. Sara J Weaver for useful discussions and Dr. Lilien Voong for critical reading of the manuscript. We also thank Dr. Songye Chen and Dr. Andrey Malyutin of the Beckman Institute Resource Center for Transmission Electron Microscopy at Caltech for assistance with data collection. This research utilized instrumentation made available by the Caltech CCE Multi-user Mass Spectrometry Laboratory, the CCE Liquids NMR Facility, and the Beckman Institute cryo-EM facility. NGS was supported by the Postdoctoral Enrichment Program from the Burroughs Wellcome Fund and DCR is a Howard Hughes Medical Institute Investigator.

## Additional information

### Funding

| Funder | Author |
| --- | --- |
| Burroughs Wellcome Fund | Naima G Sharaf |
| Howard Hughes Medical Institute | Douglas C Rees |

The funders had no role in study design, data collection and interpretation, or the decision to submit the work for publication.

### Author contributions

Naima G Sharaf, Conceptualization, Data curation, Formal analysis, Supervision, Funding acquisition, Validation, Investigation, Visualization, Methodology, Writing - original draft, Project administration, Writing - review and editing; Mona Shahgholi, Resources, Formal analysis, Investigation, Methodology, Writing - original draft, Writing - review and editing; Esther Kim, Validation, Investigation, Methodology; Jeffrey Y Lai, Software, Methodology; David G VanderVelde, Resources, Methodology; Allen T Lee, Resources, Investigation, Writing - review and editing; Douglas C Rees, Conceptualization, Data curation, Supervision, Funding acquisition, Project administration, Writing - review and editing

### Author ORCIDs

Naima G Sharaf https://orcid.org/0000-0002-3662-9228
David G VanderVelde http://orcid.org/0000-0002-2907-0366
Douglas C Rees https://orcid.org/0000-0003-4073-1185

Decision letter and Author response
Decision letter https://doi.org/10.7554/eLife.69742.sa1
Author response https://doi.org/10.7554/eLife.69742.sa2

## Additional files

### Supplementary files
• Transparent reporting form

### Data availability

For NmMetNI in the inward-facing conformation and lipo-NmMetQ:NmMetNI complex in the outward-facing conformation, cryoEM maps have been deposited in the Electron Microscopy Data Bank (EMDB) under accession codes EMD-23752 and EMD-23751. Coordinates for the model are deposited in the Research Collaboratory for Structural Bioinformatics Protein Data Bank under accession numbers 7MC0 and 7MBZ, respectively.

The following datasets were generated:

| Author(s) | Year | Dataset title | Dataset URL | Database and Identifier |
|---|---|---|---|---|
| Sharaf NG, Rees DC | 2021 | Inward facing conformation of the MetNI methionine ABC transporter | https://www.emdatare-source.org/EMD-23752 | EMDataResource, EMD-23752 |
| Sharaf NG, Rees DC | 2021 | Outward facing conformation of the MetNI methionine ABC transporter in complex with lipo-MetQ | https://www.emdatare-source.org/EMD-23751 | EMDataResource, EMD-23751 |
| Sharaf NG, Rees DC | 2021 | Outward facing conformation of the MetNI methionine ABC transporter in complex with lipo-MetQ | https://www.rcsb.org/structure/7MBZ | RCSB Protein Data Bank , 7MBZ |
| Sharaf NG, Rees DC | 2021 | Inward facing conformation of the MetNI methionine ABC transporter | https://www.rcsb.org/structure/7MC0 | RCSB Protein Data Bank, 7MC0 |

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
