## [Decision Letter]

**Acceptance summary:**

Sharaf and colleagues present an elegant structural and functional analysis of the Neisseria meningitidis substrate binding protein MetQ and the ABC transporter MetNI, demonstrating that MetQ requires N-terminal lipidation and substrate in order to stimulate ATPase activity of the transporter. This work presents MetQ as a true dual function lipoprotein present in both the inner and outer membranes. This study will be of broad interest to microbiologists, as well as those studying infectious disease and mechanisms of membrane transport.

**Decision letter after peer review:**

Thank you for submitting your article "Characterization of the ABC methionine transporter from *Neisseria meningitidis* reveals that MetQ is a lipoprotein" for consideration by *eLife*. Your article has been reviewed by 3 peer reviewers, one of whom is a member of our Board of Reviewing Editors, and the evaluation has been overseen by Kenton Swartz as the Senior Editor. The following individual involved in review of your submission has agreed to reveal their identity: Trevor Moraes (Reviewer #3).

Essential revisions:

There was general enthusiasm for this paper, with the reviewers finding that the stimulation of ATPase activity of NMetNI with the lipidated NMetQ is an interesting and important discovery. However, to strengthen the impact of the result, the following revisions need to be addressed.

1) The low resolution of the NmMetQ/NmMetNI structure is a limitation of this manuscript. While it is understandable that there are many factors that may limit the resolution, if it is possible to improve the resolution of this structure with additional imaging, then this should be included. This may address the role of lipid modifications for ATPase stimulation and strengthen the paper substantially.

2) The novelty of this finding that NmMetQ is a lipoprotein is somewhat diminished due to EcMetQ being experimentally verified to be a lipoprotein previously, as referenced on Page 3, lines 75-76 (Tokuda et al., 2007; Carlson et al., 2018). But the literature cited here does not seem accurate, in particular the Carlson et al., reference. How was EcMetQ was verified to be a lipoprotein? This background should be expanded to understand the differences between EcMetQ and NmMetQ, which may clarify the novelty of this finding in NmMetQ.

3) Given the major finding is that it is the lipidation + substrate that activates ATPase activity of the transporter, the title could be changed as it should reflect that the characterization of the ABC methionine transporter from Neisseria reveals that lipidated MetQ is required for interaction.

4) Please respond to the Recommendations to Authors below.

*Reviewer #1 (Recommendations for the authors):*

1. Regarding the interpretation of the DLS results in Figure 1 —figure supplement 1 – in the cartoons depicted, it seems that the DDM micelle is being disregarded, and that the aggregation is due to the proteins alone. Is the micelle component being subtracted in these measurements? Even if the free micelle component was being subtracted, it is possible that the micelle size changes in the presence of the different protein species. With this, these cartoons may be misleading in suggesting that the aggregates are from protein alone. In this case, these cartoons should be removed or clarified that different mixtures or protein and micelles could account for these larger sizes.

2. In Figure 2A – it is unclear what the inset graph is depicting and there is no description of this in the legends. Please clarify this information and what is the meaning of the colors on this plot.

3. Figure 1B and C – perhaps this is a matter of convention, but the y-axis labeled as "UV (mAU)" implies that one is measuring the UV signal, but rather the measurement is of the Absorbance of the UV wavelength. Therefore, "Absorbance at 280 nm (mAU)" or "A280 (mAU)" is more appropriate.

4. This request is not required, but I am left wondering how these proteins are to participate in the adhesion mechanism mentioned in the introduction and discussion. Since this paper presents such an illuminating picture of what this protein may be doing, it would be informative to elaborate on whether there is any information about the adhesion mechanisms known (is it tails up, protein-protein … ?).

5. Clarify the term "dual-topology" on Page 13, line 430. Is this referring to the dual localization of the protein in the IM and OM? In this case, it should be dual localization.

*Reviewer #2 (Recommendations for the authors):*

Please proofread the text carefully. There are several mistakes, e.g., line 68-69: "Kadner (1974,1977)"; line 308: Figure 2 should be Figure 3D; line 361: "EcNmMetQ"; ….

*Reviewer #3 (Recommendations for the authors):*

1. Figure 1, Supplementary Figure 1

The difference observed in SEC for the NmMetQ with lipoprotein, preprotein and mature would be better analyzed by SEC couple to MALS where A280 could help determine the protein vs micelle contribution to scattering. This could give a better approximation of Mw and the number of molecules in the aggregate.

2. It would be also helpful to list the type of SEC column from GE (resin, S200 INCREASE? or S75)?

3. Could there be a proportion of MetQ that is lipidated and not lipidated in Neisseria… The Mass Spec quantification would be a nice analysis tool if someone could IP MetQ from Neisseria.

4. As mentioned above, the inward facing structure adds to our understanding of how interactions between the NBDs of ABC transporters without a C2 autoinhibitory domain differ from their homologous structures with autoinhibitory domains -although this could have been described for people outside the field more clearly with a schematic figure….

5. The distance between these domains precludes the need for an autoinhibitory domain which prevent ATP turnover until SBP is bound?

6. For the FAXS experiment.. the methods section suggests that the reporter molecule is far in excess of the MetQ protein. If this were true, are you measuring the free F* (as the figure suggests) or the bound form? And would this be better informed with a titration of competitor?

---

## [Author Response]

Essential revisions:There was general enthusiasm for this paper, with the reviewers finding that the stimulation of ATPase activity of NMetNI with the lipidated NMetQ is an interesting and important discovery. However, to strengthen the impact of the result, the following revisions need to be addressed.1) The low resolution of the NmMetQ/NmMetNI structure is a limitation of this manuscript. While it is understandable that there are many factors that may limit the resolution, if it is possible to improve the resolution of this structure with additional imaging, then this should be included. This may address the role of lipid modifications for ATPase stimulation and strengthen the paper substantially.

While we very much appreciate the value of a higher resolution structure of lipo-NmMetQ:NmMetNI, we have not been able to accomplish this despite many, many attempts to optimize the sample, grid conditions and instrument settings. At this point, we do not see any obvious (or even not so obvious) path forward to improve the resolution. We would also like to highlight a recent study by Liu et al., 2020 (see below for full reference) where three structures of a mycobacterial lipoprotein-SBP:ABC transporter complex were determined in different conditions at resolutions of 3.30, 3.44 and 3.78 Å. Despite the higher resolutions, no lipids were resolved in any of these structures. We are therefore not confident that a higher resolution structure of lipo-NmMetQ:NmMetNI will yield any more detailed information on how lipidation stimulates the ATPase activity.

Liu F, Liang J, Zhang B, Gao Y, Yang X, Hu T, Yang H, Xu W, Guddat LW, Rao Z. Structural basis of trehalose recycling by the ABC transporter LpqY-SugABC. Science advances. 2020 Oct 1;6(44):eabb9833.

2) The novelty of this finding that NmMetQ is a lipoprotein is somewhat diminished due to EcMetQ being experimentally verified to be a lipoprotein previously, as referenced on Page 3, lines 75-76 (Tokuda et al., 2007; Carlson et al., 2018). But the literature cited here does not seem accurate, in particular the Carlson et al., reference. How was EcMetQ was verified to be a lipoprotein? This background should be expanded to understand the differences between EcMetQ and NmMetQ, which may clarify the novelty of this finding in NmMetQ.

We thank the reviewers for their careful reading of our paper and for finding our citation error. We have updated the manuscript with the correct Carlson et al., citation and modified the background to include how EcMetQ was verified as a lipoprotein.

3) Given the major finding is that it is the lipidation + substrate that activates ATPase activity of the transporter, the title could be changed as it should reflect that the characterization of the ABC methionine transporter from Neisseria reveals that lipidated MetQ is required for interaction.

We agree with the reviewers and have changed the title as suggested.

4) Please respond to the Recommendations to Authors below.

Please see inline text responses.

Reviewer #1 (Recommendations for the authors):1. Regarding the interpretation of the DLS results in Figure 1 —figure supplement 1 – in the cartoons depicted, it seems that the DDM micelle is being disregarded, and that the aggregation is due to the proteins alone. Is the micelle component being subtracted in these measurements? Even if the free micelle component was being subtracted, it is possible that the micelle size changes in the presence of the different protein species. With this, these cartoons may be misleading in suggesting that the aggregates are from protein alone. In this case, these cartoons should be removed or clarified that different mixtures or protein and micelles could account for these larger sizes.

Thank you. We agree that different mixtures of proteins and DDM micelles could account for these larger sizes. The cartoons have been removed and the text has been rephrased to be more cautious in describing the protein-micelle aggregates. (Line 184)

2. In Figure 2A – it is unclear what the inset graph is depicting and there is no description of this in the legends. Please clarify this information and what is the meaning of the colors on this plot.

Thank you again. This was an unintended omission. The Figure 2 caption has been updated with the description of the insert graph.

3. Figure 1B and C – perhaps this is a matter of convention, but the y-axis labeled as "UV (mAU)" implies that one is measuring the UV signal, but rather the measurement is of the Absorbance of the UV wavelength. Therefore, "Absorbance at 280 nm (mAU)" or "A280 (mAU)" is more appropriate.

Thank you – we have updated the Y-axis as suggested.

4. This request is not required, but I am left wondering how these proteins are to participate in the adhesion mechanism mentioned in the introduction and discussion. Since this paper presents such an illuminating picture of what this protein may be doing, it would be informative to elaborate on whether there is any information about the adhesion mechanisms known (is it tails up, protein-protein … ?).

This is an important question; however, to our knowledge the mechanism of how SBPs act as adhesins is not understood (including whether the adhesin activity utilizes the same substrate binding site needed for transport).

5. Clarify the term "dual-topology" on Page 13, line 430. Is this referring to the dual localization of the protein in the IM and OM? In this case, it should be dual localization.

We agree with the reviewer and have changed the term “dual topology” to “dual localization” throughout the manuscript.

Reviewer #2 (Recommendations for the authors):Please proofread the text carefully. There are several mistakes, e.g., line 68-69: "Kadner (1974,1977)"; line 308: Figure 2 should be Figure 3D; line 361: "EcNmMetQ"; ….

We thank the reviewer for finding the figure typo and have made the appropriate correction.

The format for Kadner citations is a direct result of the *eLife* Latex template. We are comfortable with *eLife* editors making this citation format change to be more consistent with the rest of the manuscript.

Reviewer #3 (Recommendations for the authors):1. Figure 1, Supplementary Figure 1The difference observed in SEC for the NmMetQ with lipoprotein, preprotein and mature would be better analyzed by SEC couple to MALS where A280 could help determine the protein vs micelle contribution to scattering. This could give a better approximation of Mw and the number of molecules in the aggregate.

We thank the reviewer for this suggestion. We used SEC-MALS to analyze NmMetQ proteins and NmMetNI as a control. Our data suggest that both pre-protein NmMetQ and lipo-NmMetQ aggregate; However, secreted NmMetQ does not. A comparison of the SEC-MALS and DLS data also revealed that pre-protein and lipo-NmMetQ molar masses measured using DLS were higher than those measured using SEC-MALS, showing that the protein-DDM aggregate size is dependent on the experimental conditions, possibly protein concentration as previously described for other lipoproteins (see reference below).

Luo Y, Friese OV, Runnels HA, Khandke L, Zlotnick G, Aulabaugh A, Gore T, Vidunas E, Raso SW, Novikova E, Byrne E. The dual role of lipids of the lipoproteins in Trumenba, a self-adjuvanting vaccine against meningococcal meningitis B disease. The AAPS journal. 2016 Nov;18(6):1562-75.

2. It would be also helpful to list the type of SEC column from GE (resin, S200 INCREASE? or S75)?

Thank you – we have added the column information to the manuscript (in the result and Materials and methods sections).

3. Could there be a proportion of MetQ that is lipidated and not lipidated in Neisseria… The Mass Spec quantification would be a nice analysis tool if someone could IP MetQ from Neisseria.

This is an interesting question. We know that non lipidated NmMetQ must be present at some concentration before it matures. The fraction of endogenous immature and mature lipo-NmMetQ however is unknown. We agree that this is a question worth pursuing in *Neisseria meningitidis*, but is beyond the scope of this work.

4. As mentioned above, the inward facing structure adds to our understanding of how interactions between the NBDs of ABC transporters without a C2 autoinhibitory domain differ from their homologous structures with autoinhibitory domains -although this could have been described for people outside the field more clearly with a schematic figure….

We thank the reviewer for this suggestion. We have added a schematic summary to Figure 4—figure supplement 2.

5. The distance between these domains precludes the need for an autoinhibitory domain which prevent ATP turnover until SBP is bound?

Comparisons of various ABC importers (Figure 4 —figure supplement 2) demonstrates that the separation between the nucleotide binding domains (NBDs) is influenced by the presence or absence of autoinhibitory domains. The connection between this observation and the basal ATPase rate is not clear, however. The autoinhibitory domain represents a regulatory mechanism that limits transport under conditions of high intracellular concentrations of methionine. It essentially functions as a two-state switch that either permits ATPase activity (low methionine conc) or prevents it (high methionine conc). Under the low methionine concentrations corresponding to our experimental conditions, the autoinhibitory domains should be in the "on" state, so that both EcMetNI and NmMetNI should be able to hydrolyze ATP. Experimentally, we find that NmMetNI is more tightly coupled than EcMetNI. It is tempting to attribute the higher basal activity of EcMetNI to the tethering together of the NBDs by the autoinhibitory domains, allowing them to come together more readily than the untethered NBDs in NmMetNI. While plausible, there is no conclusive experimental data to support or refute this model.

6. For the FAXS experiment.. the methods section suggests that the reporter molecule is far in excess of the MetQ protein. If this were true, are you measuring the free F* (as the figure suggests) or the bound form? And would this be better informed with a titration of competitor?

An important aspect of the FAXS experiment is the requirement of a weak affinity reporter molecule. For weak affinity ligands in fast exchange on the NMR chemical shift timescale, a single resonance is observed whose population is the weighted average of the free and bound states. Therefore, the displacement of the reporter molecule by the competing methionine analog results in the increase of signal intensity due to an increase of the fraction of free reporter molecules and an accompanying decrease in the fraction of reporter molecules bound to NmMetQ. The overall signal intensity reflects the weighted average of both the free and bound populations of the reporter molecule.We agree that this experiment could be performed at different concentrations of competing ligands, but to obtain the relative binding affinities of the methionine analogs, only one condition is needed.